# Preparation of Porous Liquid Based on Silicalite-1

**DOI:** 10.3390/ma12233984

**Published:** 2019-12-01

**Authors:** Yutong Liu, Yang Bai, Tao Tian

**Affiliations:** 1Key Laboratory of Groundwater Resources and Environment, Ministry of Education, College of Environment and Resource, Jilin University, Changchun 130012, China; liuyt841011@163.com (Y.L.); 15943013272@163.com (Y.B.); 2Jilin Engineering Normal University, Changchun 130012, China

**Keywords:** porous liquid, silicalite-1, CO_2_ capacities, gas storage

## Abstract

Solid porous materials, like zeolites, have been widely used in a variety of fields such as size-and-shape-selective absorption/separation and catalysis because of their porosity. However, there are few liquid materials that exhibit permanent porosity. Porous liquids are a novel material that combine the properties of fluidity and permanent porosity. They have potential applications in many fields such as gas separation, storage and transport. Herein, we report a novel Type 1 porous liquid prepared based on silicalite-1. The pore size of this porous liquid was determined by positron annihilation lifetime spectroscopy (PALS), and the CO_2_ capacities were determined by the intelligent gravimetric analyzer (IGA). The unique properties of this porous liquid can promote its application in many fields such as gas storage and transport.

## 1. Introduction

Zeolites, a family of microporous aluminosilicates characterized by a regular system of inner channels and uniform open pores [1,2], have been widely used in size-and-shape-selective absorption/separation [3,4,5,6,7,8,9,10,11] and catalysis [12,13,14,15] due to their sharp pore size distribution, large surface area and thermal/hydrothermal stability. When zeolites are used as adsorbents, they have high adsorption capacity and selectivity for a variety of gases such as He [4], CO_2_ [7], O_2_ [10], N_2_, CH_4_, C_2_H_6_ [11] and so on.

As solid adsorbents, they have many advantages, such as lower energy penalties in adsorption-desorption cycles, but they are difficult to apply in conventional liquid processes because of their solid properties [16,17]. One way to solve this problem is by liquefiying the zeolites into porous liquids that combine the properties of fluidity and permanent porosity.

There are three types of porous liquids proposed by James et al. based on the nature of the host systems [16]. Type 1 is neat liquid hosts that cannot collapse or interpenetrate; type 2 and 3 are rigid hosts or particles of microporous frameworks dispersed in sterically hindered solvents, respectively. By now, several porous liquids have been reported. Dai’s group had reported a Type 1 porous liquid using hollow silica spheres as hosts (denoted by OS@HS) and determined the gas separation for N_2_/CO_2_ [18]. Although the gas permeability and selectivity are moderately low, the good tunability of this hybrid system affords ample choices for further optimization. James and co-workers had prepared “porous liquids” by dissolving rigid organic cage molecules into a sterically hindered solvent; the solubility of methane gas in this liquid increased about 8 times [17]. We have reported a class of Type 3 porous liquids synthesized by solution mixing of porous hosts (ZIF-8, ZSM-5 and Silicalite-1) and rationally designed ionic liquids [19]. The CO_2_ capacity measurements confirm the permanent porosity in these liquids. These liquids have potential applications in many fields such as gas separation, storage and transport. 

Herein, a Type 1 porous liquid based on nanosized silicalite-1 (S-1) zeolite, denoted as S-1-Liquid, has been successfully prepared. The compound was characterized by Fourier-transform infrared (FT-IR), thermogravimetric analysis (TGA), X-ray diffraction (XRD), N_2_ adsorption/desorption, scanning electron microscopy (SEM) and transmission electron microscopy (TEM). The positron annihilation lifetime spectroscopy (PALS) and CO_2_ capacities determined by the intelligent gravimetric analyzer (IGA) were used to demonstrate its porosity. 

## 2. Materials and Methods 

The Type 1 porous liquid (S-1-Liquid) was prepared in a three-step synthetic procedure, as shown in Scheme 1.

### 2.1. Synthesis of Nanosized Silicalite-1

Nanosized S-1 zeolite has been synthesized under direct hydrothermal conditions [20]. The initial solutions were prepared by mixing tetraethyl orthosilicate (98%, Aldrich, Shanghai, China) and tetrapropylammonium hydroxide (1 M aqueous solution, Aldrich, Shanghai, China). The chemical composition is 9 TPAOH: 25 SiO_2_: 480 H_2_O. The silica alkoxides were hydrolyzed under slow stirring (50 rpm) for 24 h at room temperature. The obtained clear solutions were transferred in polypropylene bottles and subjected to hydrothermal treatment in a conventional oven. The syntheses were performed at 90 °C for 96 h. After the treatment, the zeolite suspensions were purified in a series of four steps consisting of high-speed centrifugation (10,000 rpm, 20 min), removal of the mother liquor, and washed by water for several times. After that, the sample was dried at 80 °C in the oven overnight and calcined at 600 °C for 6 h to remove organic impurity.

### 2.2. Surface Sol-gel Process (SSP) on Silicalite-1

The SSP on S-1 surface was based on literature reports [21]. Typically, 1.0 g of pre-dried S-1 zeolite powder was added into a reactor under an anhydrous condition. Subsequently, 5 mL of titanium(IV) butoxide (97%, Aldrich, Shanghai, China) in a mixture of 10 mL of anhydrous toluene (99.8%, Aldrich, Shanghai, China) and 10 mL of anhydrous methanol (99.8%, Aldrich, Shanghai, China) was added into the reactor through a syringe. After 30 min, the precipitate was separated and washed with anhydrous methanol to remove the unreacted titanium(IV) butoxide. After that, an excess amount of water was mixed with the resulting sample to hydrolyze the monolayer of titanium oxide. Finally, the sample was washed with anhydrous methanol several times and dried at 40 °C in a vacuum oven.

### 2.3. Preparation of S-1-Liquid

A total of 1.0 g of S-1 was dispersed in 20 mL deionized water and sonicated for 10 min. Then, 2.0 mL of organosilane (OS), (CH_3_O)_3_Si(CH_2_)_3_N^+^(CH_3_)(C_10_H_21_)_2_Cl (Aldrich, Shanghai, China), was added into the suspension under vigorous stirring. Subsequently, the mixture was aged at room temperature for 24 h. After that, the precipitate, denoted as OS@S-1, was washed with water for three times and with ethanol for three times, respectively, and dried at 100 °C overnight. Finally, the OS@S-1 was treated by 15.0 mL poly(ethylene glycol) tailed sulfonate (PEGS, 16.5%, C_9_H_19_-C_6_H_4_-(OCH_2_CH_2_)_20_O-(CH_2_)_3_SO_3_K^+^, Aldrich, Shanghai, China) at 70 °C for 24 h. The mixture was dried at 70 °C and dispersed in 15.0 mL acetone for three times to remove the KCl byproduct. The sample was dried at 70 °C for 24 h to obtain the S-1-Liquid. The S-1-Liquid was kept at 40 °C under vacuum for use. 

In the beginning, nanosized S-1 zeolite was used directly to prepare the S-1-Liquid, as shown in Scheme 1a. However, disappointingly, after the replace of chloride anion of the OS by PEGS, we cannot obtain a stable liquid phase. It is probably because there are too few hydroxy groups on the surface of the S-1 zeolite. Then, we changed the strategy. The surface sol-gel process (SSP) was used to increase the surface hydroxy groups of S-1 zeolite before the surface modification by OS, as shown in Scheme 1b, yielding an optically transparent zeolite-based porous liquid (S-1-Liquid) at room temperature.

## 3. Results and Discussion

The Fourier-transform infrared (FT-IR) spectra of products in each step are presented in Figure 1a. For the nanosized S-1 zeolite, the absorption peak at 799 cm^−1^ is due to the symmetric stretching vibrations of the Si–O–Si bonds of zeolite and the bands at 1074 cm^−1^ and 1225 cm^−1^ are attributed to the asymmetric stretching vibrations of the Si–O–Si bonds. The IR spectrum of S-1 after SSP is closely related to that of S-1 (Appendix A). After surface modification, some featured bands assigned to the OS moiety are clearly visible in the OS@S-1 spectrum, such as the stretching and bending vibrations of –CH_2_– backbones at 2928 cm^−1^, 2870 cm^−1^ and 1470 cm^−1^. The bands at 799 cm^−1^, 1074 cm^−1^ and 1225 cm^−1^, which are attributed to the stretching vibrations of the Si–O–Si bonds of zeolite, are further enhanced. In the spectrum of the S-1-Liquid, additional bands such as 2868 cm^−1^, 1724 cm^−1^, 1186 cm^−1^ and 1094 cm^−1^ are attributed to the aliphatic, phenyl, sulfonate and ether, respectively. It means that the OS@S-1 is surrounded by the PEGS completely to obtain the S-1-Liquid. The S-1 after SSP was examined by XPS to determine the existence of titanium species on the surface of S-1. Appendix A shows the XPS spectrum on the P_2_p region of the S-1 after SSP; two emission peaks are observed at 458.0 eV and 464.0 eV for Ti 2p_3/2_ and Ti 2p_1/2_, respectively, indicating the existence of titanium in the Ti^4+^ state on the surface of S-1.

Figure 1b shows the thermogravimetric analysis (TGA) of S-1, OS@S-1, and S-1-Liquid. There is about 2.7% of mass loss during heating until the temperature reaches 700 °C for the S-1, according to the removal of the adsorption water. The S-1 after treated by SSP has a mass loss of about 2.8% (Appendix A), similar to the S-1, which means that there are no organic group residue on S-1 surface after SSP. For the OS@S-1, the loss weight is about 13.3%, corresponding to the decomposition of OS. In comparison, when the S-1 before treated by SSP reacts with the OS, the mass loss is about 6.8%, much lower than OS@S-1 (Appendix A). This demonstrates that the S-1 after SSP has more surface hydroxyl compared to the S-1 to react with the OS. It means that the surface sol-gel process is efficient to increase the surface hydroxyl of S-1. There is no significant mass loss during heating until the temperature reaches approximately 190 °C, demonstrating that the S-1-Liquid is solvent-free. From 190 °C to 460 °C, there is 67% of mass loss for S-1-Liquid, corresponding to the decomposition of organic groups around the S-1 surfaces. The final loss weight is 66.5% when the temperature approaches 700 °C. 

As shown in Figure 2a, the X-ray diffraction pattern of S-1 shows a typical characteristic diffraction pattern of the MFI-type zeolite. The four peaks at 2θ of 8.08, 9.04, 23.3 and 24.1° corresponded to the (101), (200), (501) and (303) planes respectively [3]. The S-1-Liquid has a similar XRD pattern with S-1, indicating the structure of S-1 has been preserved in the liquid. To confirm that the pores of S-1 zeolite remained empty after OS modification, OS@S-1 was also characterized by N_2_ adsorption/desorption isotherms (Figure 2b). There was no significant change in the N_2_ adsorption/desorption isotherm for OS@S-1, compared to S-1 zeolite. It means that the pores of S-1 zeolite remained empty after surface modification by OS. This is an important result to demonstrate that the OS@S-1 still has empty pores for gas transport or storage.

The as-synthesized nanosized S-1 and OS@S-1 were determined by transmission electron microscopy (TEM), as shown in Appendix A. The crystal size of S-1 is around 100 nm, and the size distribution of OS@S-1 is about 150 nm to 200 nm, which is larger than that of S-1 because of the surface modification by OS. The nanostructure of S-1-Liquid was imaged by scanning electron microscopy (SEM) and (TEM). As observed in Figure 3, the S-1 nanocrystals are defined well in the PEGS to form a liquid-like polymeric medium. These nanocrystals are combined together to form large agglomerates through the strong interaction among the surface functional groups. 

Positron annihilation lifetime spectroscopy (PALS) was then used to determine the average size of the pores contained within the studied S-1-Liquid. The PALS technique measures the lifetime and intensity of positions that annihilate in materials when the material of interest is exposed to a positron source, which is ^22^NaCl in this study. Once entering the materials, positrons experience thermalization, diffusion, and finally are trapped and annihilated by electrons. The positron lifetime is dependent on the overlap of the positron wavefunction with the wavefunctions of the electrons in materials. Therefore, the electron density distribution in materials, in particular, the presence of open volume, can be obtained by using this technique. The positron lifetime is defined as the time interval between the injection of positions into the materials (indicated by a birth gamma ray of 1.274 MeV) and the decay of the positron-electron pairs (indicated by two 0.511 MeV gamma rays traveling in opposite directions). Data were collected using a digital oscilloscope with a system timing resolution of 173 ps. More details about the PALS system could be found in Reference [22].

Figure 4 shows the recorded positron lifetime spectrum as well as the fitting curve by applying 3-component analysis. The first lifetime (τ1) indicates the positrons being annihilated in the bulk of the materials (<200 ps). The second lifetime, τ2, is attributed to the positrons being annihilated in defects (300–500 ps). The third lifetime, τ3, refers to ortho-positronium (o-Ps), a parallel spin complex of a positron and electron, which forms in low electron density regions, such as free volumes, holes, interfaces, and pores. The o-Ps lifetime and intensity are often associated with the size and concentration, respectively, of the open volume in materials. From the fitting procedure one obtains positron lifetimes and intensities. The o-Ps lifetime, τ3, is typically related to the average radius of a free volume element, r, which is assumed to be spherical, by Tao-Eldrup model,
(1)$$\tau_{3}=\frac{1}{2}{\left(1-\frac{r}{r+\Delta r}+\frac{1}{2\pi }\sin\left[2\pi \frac{r}{r+\Delta r}\right]\right)}^{-1}$$
where Δr is the empirical electron layer thickness, which is taken to be 1.66 A [23,24].

By applying Equation (1), the o-Ps lifetime of 2.251 ns correlates to an average pore size of 0.61 nm in the S-1-Liquid; this is consistent with the pore size of S-1 zeolite.

To further prove that the pores of S-1 are still empty in the S-1-Liquid, the CO_2_ capacities of it were determined by an Intelligent Gravimetric Analyzer. The CO_2_ uptake capacity of S-1-Liquid is enhanced compared to the PEGS, as shown in Figure 5. The CO_2_ uptake value of S-1-Liquid is 0.474 wt % at the pressure of 1 bar, while the uptake value of PEGS is 0.261 wt %. When the pressure increases to 10 bar, the CO_2_ uptake value of S-1-Liquid is 2.524 wt %, which remains higher than that of PEGS (2.261 wt %). Both the S-1-Liquid and PEGS have hysteresis loops. The S-1-Liquid remains more CO_2_ as the pressure decreases during the desorption process due to the permanent porosity. The CO_2_ uptake value of S-1-Liquid is 1.406 wt % at 1 bar, much higher than that of PEGS (0.809 wt %). There is 55.7% wt % of CO_2_ remains in the S-1-Liquid, demonstrating the potential application for CO_2_ storage.

## 4. Conclusions

In summary, a Type 1 porous liquid S-1-Liquid was synthesized in a three-step synthetic procedure. This porous liquid is stable and homogeneous at room temperature. It exhibits enhanced CO_2_ adsorption capacity due to the permanent porosity, which was also confirmed by PALS results. We believe that this porous liquid has potential applications in many fields such as gas storage and transport.

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
