# Peer review of "Preparation of Porous Liquid Based on Silicalite-1"

_materials, 2019, doi:10.3390/ma12233984_

Round 1

Reviewer 1 Report

The manuscript entitled “ Preparation of Porous Liquid based on Silicalite-1” reported the preparation of  Type 1 porous liquid prepared based on Silicalite-1. The authors determined the pore size of this porous liquid by positron annihilation life time spectroscopy (PALS), and the CO2 capacities by the intelligent gravimetric analyser.

The article is interesting, the literature review is sufficient.

I recommend this manuscript to publication with major revision.

My comments are following:

The authors presented the S-1 liquid characteristic but they practically did not show the characteristics of nanosized silicalite-1 unmodified and modified by titanium diooxide.

Authors should add a results characterizing nanosized silicalite-1: TEM images, crystal size distributions.

Line 119 -120 – after the sentence: “As shown in Figure 2a, the X-ray diffraction patterns of S-1 and S-1-Liquid show the similar peaks characteristic of the MFI-type zeolite, indicating the structure of S-1 has been preserved in the liquid.” a reference confirming the interpretation of the diffraction pattern of S-1 should be given. Authors should analyze a diffraction pattern of S-1 to prove that this pattern corresponds to the MFI-type zeolite.

2.2. Surface sol-gel process (SSP) on Silicalite-1

“After that, an excess amount of water was mixed with the resulting sample to hydrolyze the monolayer of titanium oxide.” I found no evidence in the text that the nanoparticles of Silicalite-1 were covered with a TiO2 monolayer. Scheme 1. The explanations of abbreviations used should be included in the description of the scheme.Figure 1. The bands should be marked in the figure.

Author Response

Dear Reviewer,

We have studied the valuable comments from you carefully and tried our best to revise the manuscript. The point to point responses are listed as following:

Point 1: The authors presented the S-1 liquid characteristic but they practically did not show the characteristics of nanosized silicalite-1 unmodified and modified by titanium dioxide.

Authors should add a result characterizing nanosized silicalite-1: TEM images, crystal size distributions.

Response 1: Thank you for your valuable advice. We have added the TEM images of nanosized silicalite-1 and OS@S-1 in the supporting information as Figure S3 and described it in the manuscript from line 138 to 141.

Point 2: Line 119 -120 after the sentence: “As shown in Figure 2a, the X-ray diffraction patterns of S-1 and S-1-Liquid show the similar peaks characteristic of the MFI-type zeolite, indicating the structure of S-1 has been preserved in the liquid.” a reference confirming the interpretation of the diffraction pattern of S-1 should be given. Authors should analyze a diffraction pattern of S-1 to prove that this pattern corresponds to the MFI-type zeolite.

Response 2: Thank you for your instructive suggestions. We have analyzed the XRD pattern of S-1 in the manuscript as shown inline 124 to 127, and a reference according to it is given.

Point 3: 2.2. Surface sol-gel process (SSP) on Silicalite-1 “After that, an excess amount of water was mixed with the resulting sample to hydrolyze the monolayer of titanium oxide.” I found no evidence in the text that the nanoparticles of Silicalite-1 were covered with a TiO2 monolayer.

Response 3: Thank you for your careful work. The S-1 after SSP was examined by XPS to determine the existence of titanium species on the surface of S-1, as shown in Figure S2. It is described in the manuscript from line 104 to 108.

Point 4: Scheme 1.The explanations of abbreviations used should be included in the description of the scheme.

Response 4: Thank you for your careful work. The explanations of abbreviations were added to the scheme.

Point 5: Figure 1. The bands should be marked in the figure.

Response5: Thank you for your valuable advice. We have marked the bands in Figure 1.

Reviewer 2 Report

The authors prepared porous silicate liquid for gas storage applications. The experimental results are interesting however more characterization results needed to confirm the porous structure of the liquid. 

Figure quality must be improved in the revised manuscript. The showed morphological images are failed to confirm the porous structure. High-magnification images should be provided for porous liquid SEM and TEM. Typo, Page 6, line 175, "Type 3 porous liquid" In Figure 2a. Why XRD peaks are shifted in S-1 Liquid compare to S-1?

Author Response

Dear Reviewer,

We have studied the valuable comments from you carefully and tried our best to revise the manuscript. The point to point responses are listed as following:

Point 1: Figure quality must be improved in the revised manuscript.

Response 1: Thank you for your valuable advice. Figure quality was improved in the manuscript.

Point 2: The showed morphological images are failed to confirm the porous structure. High-magnification images should be provided for porous liquid SEM and TEM.

Response 2: Thank you for your instructive suggestions. We have tried to obtain high-magnification images of porous liquid, but unfortunately, we failed. This may be because the organic matter coating on S-1 blocks the electron beam from penetrating. To make up for this deficiency, positron annihilation lifetime spectroscopy (PALS) was used to determine the average size of the pores contained within the studied S-1-Liquid.

Point 3: Typo, Page 6, line 175, "Type 3 porous liquid"

Response 3: Thank you for your careful work. It should be “Type 1 porous liquid”.

Point 4: Why XRD peaks in S-1 Liquid compare to S-1?

Response 4: Thank you for your careful work. we are sorry for the carelessness that the shift might be owing to the system error. The coating of organic matters upon S-1 would not affect the crystallization and diffraction. In the revised paper, it has been redetermined.

Round 2

Reviewer 1 Report

Corrections are sufficient.

Reviewer 2 Report

Authors carried out all of my suggested comments in the revised version. I would recommend this work for publication in Materials.